# Aluminum Bronze/Udimet 500 Composites Prepared by Electron-Beam Additive Double-Wire-Feed Manufacturing

**DOI:** 10.3390/ma15186270

**Published:** 2022-09-09

**Authors:** Anna Zykova, Andrey Chumaevskii, Aleksandr Panfilov, Andrey Vorontsov, Aleksandra Nikolaeva, Kseniya Osipovich, Anastasija Gusarova, Valentina Chebodaeva, Sergey Nikonov, Denis Gurianov, Andrey Filippov, Artem Dobrovolsky, Evgeny Kolubaev, Sergei Tarasov

**Affiliations:** Institute of Strength Physics and Materials Science, Siberian Branch of Russian Academy of Sciences, Tomsk 634055, Russia

**Keywords:** electron-beam additive manufacturing, in situ composite, phase transformation, dispersion hardening

## Abstract

Novel composite CuA19Mn2/Udimet-500 alloy walls with different content of the Udimet 500 were built using electron-beam double-wire-feed additive manufacturing. Intermixing both metals within the melted pool resulted in dissolving nickel and forcing out the aluminum from bronze. The resulting phases were NiAl particles and grains, M_23_C_6_/NiAl core/shell particles and Cu-Ni-Al solid solution. Precipitation of these phases resulted in the increased hardness and tensile strength as well as reduced ductility of the composite alloys. Such a hardening resulted in improving the wear resistance as compared to that of source aluminum bronze.

## 1. Introduction

The development of new composite and hybrid materials that allow combining useful characteristics of their constituents has long been a challenge in materials science. The main process used for obtaining bulk composites was powder metallurgy where microstructures were formed by intermixing and sintering the powder components. It could not be possible to interfere with the process and modify the microstructure in situ. Such an opportunity appeared only with the development of additive manufacturing when each layer may be deposited under different process parameters and using different materials. Therefore, this new principle of composite structure forming would allow obtaining different combinations of components and characteristics.

A combination of high-temperature-strength nickel-base superalloys with high-heat-conductivity copper-base alloys can be a good approach for improving the performance of components and units used in nuclear power installations and jet vehicles. The development of additive manufacturing methods may afford a convenient ground for the controlled layer-by-layer production of such material [1,2,3,4,5,6,7,8,9]. The desired most effective structures can be first modeled and then reproduced using a continuously metered feed of the components. The use of double-wire-feed additive manufacturing is the most productive approach in such a case since it allows controlling wire feed rates in situ and thus obtaining the desired combination of components.

Apart from high-temperature strength and high heat conductivity, an important functional characteristic of such a composite alloy material would be its wear resistance. Luckily, heat-conductive copper alloys are used also as slide-bearing materials to kill two birds with one stone. In particular, high-strength and corrosion-resistant aluminum bronzes with an aluminum content of 8–9 wt.% are widely used for making slide bearings for high-speed vehicles [10,11]. Aluminum bronzes with an aluminum content of >17 wt.% show high wear resistance but less tensile strength because of embrittlement from intermetallic phases such as γ-Cu_9_Al_4_ [11]_._

Nickel is the main component of high-temperature-strength superalloys and therefore could dissolve in the aluminum bronze during additive manufacturing. Nickel–aluminum bronzes are well-known alloys that have a complex microstructure composed of α-grains, a residual β-phase and four types of κ-phases [12,13]. The latter allow improving strength without losing ductility.

The formation of all these phases as well as others can be anticipated during the wire additive manufacturing of composite alloys from nickel superalloy and aluminum bronze wires.

Therefore, one of the tasks was studying the phases and microstructures that would form in composite alloys depending upon the nickel superalloy content. Another task was the evaluation of their effect on the mechanical and tribological characteristics of the materials.

## 2. Materials and Methods

Samples of composite alloys such as CuAl9Mn2/Udimet-500 were used for double-wire electron-beam additive manufacturing (Figure 1). Both wires with their element compositions as shown in Table 1 had the same diameter of ∅1.6 mm and were fed into a molten pool formed by an electron beam at a constant accelerating voltage of 30 kV. The beam current of 72 mA was used for depositing the first few layers on an AISI 321 steel to compensate for fast heat removal via the cold substrate and then was reduced to 44 mA to avoid excess heat input due to reduced heat removal. The substrate speed was kept constant at 400 m/min. This set of parameters was the result of many previous experiments on electron-beam wire-feed building walls of stainless steel, heat-resistant superalloys and composite materials. The required concentration of the Udimet-500 in the bronze was obtained by automatically adjusting and maintaining the corresponding wire-feed rates. Such an approach allowed building four multilayer walls whose compositions are shown in Table 1. For convenience, the composites containing 5, 10, 15 and 25 vol.% of Udimet-500 were denoted as 5% Ud, 10% Ud, 15% Ud and 25% Ud, respectively.

Samples (Figure 1, pos. 1) intended for studying microstructures, phases and microhardness were ground, polished, etched in a solution of 30 mL HCl + 5 g FeCl_3_-6H_2_O + 60 mL H_2_O and then examined using an Altami Met 1S (Altami Ltd., Saint-Petersburg, Russia) optical microscope and a high-resolution field emission scanning electron microscope (HR FESEM) Apreo 2 S (Thermo Fisher Scientific, Waltham, MA, USA), equipped with an Octane Elect Super (EDAX, Mahwah, NJ, USA) EDS detector and a Velocity Super (EDAX, Mahwah, NJ, USA) EBSD detector. TEM studies were carried out using a JEOL-2100, (JEOL Ltd., Tokyo, Japan) instrument.

An X-ray fluorescence spectrometer Niton XL3t 980 GOLDD (Thermo Fisher Scientific, Waltham, MA, USA) allowed determining the element composition of the materials. The phases formed in them were identified using an XRD instrument XRD-7000S, Co_Kα._ Microhardness numbers were obtained using a Vickers microhardness tester Duramin 5 (Struers A/S, Ballerup, Danemark). Mechanical testing was carried out using a tensile machine UTS-110M (Testsystems, Ivanovo, Russia) on samples cut from the walls so that their tensile axes coincided with two perpendicular directions as shown in Figure 1.

Sliding wear testing was carried out using a ball-on-disk scheme with Si_3_N_4_ ∅6 mm ball sliding on the composite alloy disk. The testing was carried out using normal load values as follows: 9, 14, 19 and 25 N. Sliding speed was 0.1 m/s. Wear was quantized by measuring the wear groove cross-section area.

## 3. Results

### 3.1. Macrostructure and XRD of the Bronze/Udimet-500

The optical microscopy images in Figure 2 allow observing macrostructures formed by intermixing and solidifying the Udimet-500 with aluminum bronze. The alloy resulting from adding Udimet-500 into aluminum bronze (5% Ud) is characterized by coarse α-Cu matrix grains and smaller dark β-Cu_3_Al ones with both grain boundary and intragrain particles (Figure 2a). Adding 10 vol.% Udimet (10% Ud) resulted in a reduced content of β-Cu_3_Al grains but an increased amount of particles. Further increasing the concentration of the alloy in the bronze to 15 vol.% (15% Ud) and 25 vol.% (25% Ud) allowed forming large grains with numerous particles that even become the grains themselves (Figure 2b–d). According to the grain size distributions in Figure 3a–d, the mean sizes of the matrix grains in the 5% Ud, 10% Ud, 15% Ud and 25% Ud composites were 41.8 ± 16.6 μm, 15.4 ± 4.7 μm, 13.9 ± 6.8 μm and 10.9 ± 7.1 μm, respectively.

XRD on all samples allowed detecting their basic phases such as α-Cu and NiAl (Figure 4). The content of NiAl increased with the percentage of Udimet-500 intermixed with bronze. Taking into account the above results, it could be suggested that intermetallic NiAl particles form in the composites which look like isolated particles in 5% Ud and 10% Ud alloys and then form coarse grains in 15% Ud and especially in the 25% Ud ones.

### 3.2. Microstructure of the Bronze/Udimet-500 Composites

SEM images of the 5% Ud sample microstructures as well as corresponding EDS element distribution maps allow identifying α-Cu grains that additionally contain Ni, Al, Mn and Co (Figure 5a,b). The Udimet-500 component is represented by coarse Cr-rich particles distributed in the α-Cu grains as well as at their boundaries. The particle size is in the range of ~300 to ~800 nm. Smaller precipitates can be observed also between the α-Cu grains (Figure 5b) that could be NiAl ones as follows from the XRD.

The size of Cr-enriched particles is increased to 1–12.5 μm with increasing the content of Udimet-500 in aluminum bronze (Figure 5c,e,g, pos.1). Along with that, these particles form agglomerates such as those shown in Figure 5e–h. TEM bright-field images of the Cr-rich particles in composites show how they grow depending on the amount of Udimet-500 introduced into the melted pool (Figure 6a–d).

An SAED pattern from the Cr-rich particle in Figure 7a,i allowed identifying only NiAl and α-Cu (Figure 7b) as confirmed by the dark-field images in Figure 7c,d obtained in reflections (002)_α-Cu_ and (110)_NiAl_. The EDS element distribution patterns show the presence of Mo in the center of the Cr-rich particle (Figure 7k,j) in the copper matrix (Figure 7J) while both Ni and Al are concentrated on the periphery of the particle. NIAl particles are clearly identified in Figure 7i,h.

All composites show the presence of spherical fine NiAl precipitates of mean size ~450 nm (Figure 5a,c,e,g and Figure 8a–d, position 2) as identified from XRD, the SAED pattern in Figure 8e and the dark-field image in Figure 8g,h. The content of these particles is increased with the content of Udimet-500 intermixed with the aluminum bronze.

Large NiAl grains were observed in the 15% Ud composite (Figure 5e and Figure 9a–c) which contained fine Cr-rich precipitates (Figure 9f). The mean size of NiAl grains in the 15% Ud composite is ~2.5 μm (Figure 5e). The TEM image in Figure 9a allows observing some fine <200 nm in size precipitates inside the NiAl grain whose EDS spectra show the presence of 25 to 49 at.% of Cu in them (Figure 9c, Table 2, spectra 1–3). For comparison, the Cu concentration in the NiAl grain is about 13 at.% (Figure 9c, Table 2, spectrum 4). It could be suggested that these copper-rich particles precipitated from the NiAl during cooling. In addition, NiAl grains contain even finer precipitates whose SAED pattern allowed identifying them as M_23_C_6_ carbides (Figure 10a,b). The dark-field images in (422)_Cr23M6_ reflection confirmed that these superfine particles precipitated from the NiAl (Figure 10c).

Admixing 25 vol.% Udimet-500 to the aluminum bronze resulted in increasing the mean size of NiAl grains up to ~90.3 μm (Figure 5g), i.e., by a factor of 3.7 larger as compared to that in the 15% Ud. According to the TEM, these NiAl grains contain fine precipitates of different morphological types and compositions (Figure 9b–l). First of all, these are Cr-rich precipitates (Figure 11e, Table 3, spectra 3,4), the same as those observed in the 15% Ud composite and Cu-rich rodlike ones (Figure 11a,c, Table 3, spectra 5,6). The latter ones contain up to 12.9 at.% of Cu.

### 3.3. Microhardness of the Composites “Bronze/Udimet-500”

Microhardness number distributions (Figure 12) allow observing a large scatter of the values that result from the microstructural inhomogeneity and indenting a relatively soft α-Cu matrix as well as NiAl and M_23_C_6_ regions. For pure bronze samples, there is a hard β-phase that gives higher microhardness numbers. The mean hardness of the composite grows with the amount of Udimet-500 intermixed with the bronze so that the mean microhardness levels of composites 5% Ud, 10% Ud, 15% Ud and 25% Ud are 1.7 ± 0.1, 1.8 ± 0.1, 2.1 ± 0.1 and 2.6 ± 0.1 Gpa, respectively.

Such a hardening can be related to solid solution hardening because of forming a Ni-Cu solid solution, NiAl particles and M_23_C_6_ carbides whose contents and sizes increase with the content of Udimet-500.

### 3.4. Mechanical Properties of the Composites “Bronze/Udimet-500”

Stress–strain tensile curves were obtained for samples oriented by their tensile axes along Y and Z (samples 3 and 4 in Figure 1) which then were averaged by three samples and reproduced as diagrams in Figure 13a,b. The concentration dependencies of ultimate tensile stress (UTS) and strain-to-fracture (STF) allow observing the tendency for increasing UTS and reducing STF as the Udimet-500 concentration grows (Figure 13c). The pure bronze after additive manufacturing has a UTS of ~480 ± 10 Mpa and an STF of ~50–60% [14]. When adding the Udimet-500, the ultimate tensile stress might increase by 75% as compared to that of the aluminum bronze. Tensile tests showed that the ultimate tensile strength magnitudes of samples oriented along the building direction Z were a little higher than those oriented along the Y direction for 5% Ud, 10% Ud and 25% Ud while the reverse behavior was observed in the example of the 15% Ud sample (Figure 13). Such a result was not typical for the additively manufactured materials with columnar grains and a corresponding anisotropy of mechanical characteristics.

SEM images of fracture surfaces demonstrate a viscous type of fracture with numerous ridges and cells (Figure 14a,c,e,g). EDS maps demonstrate the distribution of elements on the fracture surfaces. Apart from copper, the main elements detected on the fracture surfaces are Cr, Mo, Ni and Al (Figure 14b,d,f,h). Such a distribution is evidence in favor of fracture occurring by NiAl and M_23_C_6_ particles.

### 3.5. Sliding Wear Testing

Wear vs. time dependencies were obtained in the course of the tribological testing of the composites (Figure 15a) at different normal loads. Sliding wear at a 9 N load was at almost the same level for all the samples studied and then almost collectively increased at 14 N. Within the 14–19 N range, wear reduction was observed on the 15% Ud sample while all other samples continuously demonstrated wear growth up to that at a 25 N load. The most intense wear was observed for the 25% Ud sample while wear reduction occurred for the 15% Ud one starting from the 14 N load.

Coefficient of friction (CoF) values were determined on achieving the steady-sliding regime characterized by maintaining the CoF value constant. The CoF steady-sliding values changed slightly in the range 0.45–0.6 for the 10% Ud, 15% Ud and 25% Ud samples while that of 5% Ud reduced as the normal load increased from 9 to 25 N (Figure 15b).

Such behavior may be typical with bronzes because of generating a tribological layer composed of oxides and ductile bronze. Such a composite layer serves to increase the real contact area at higher loads and reduce adhesive wear component. It is rather typical behavior for bronzes in sliding contact, and therefore they are used in sliding bearing. The worn surfaces corresponding to 14 N, and 19 and 25 N normal loads look all the same despite CoF reduction from 0.45 to 0.25.

Let us note that, generally, there is no correlation between wear and the CoF depending upon the wear mechanism. The work surfaces obtained after tribological tests are represented by smooth glazed areas, rough areas and wear particles (Figure 16, Table 4). The glazed areas on the 5% Ud and 10% Ud samples contained less oxygen as compared to those of rough ones and wear particles. The amount of these glazed areas reduces as the normal load grows, but more and more wear particles are produced. An examination of worn surfaces shows that the amount of wear particles grows with the normal load. An EDS elemental analysis of both wear particles and worn surfaces demonstrated the identity of their composition. Both worn surfaces and wear particles were heavily oxidized. These results are not shown in the article for brevity.

## 4. Discussion

Although extensive experimental studies have been carried out on additive manufacturing of nickel-base superalloys using laser and electron beam melting [15,16] no attempts were made on obtaining additive manufactured supealloy/bronze composite alloys. 

Apart from powder-bed laser or electron beam processes, the most widely discussed additive process now is WAAM (wire arc additive manufacturing), where wire is fed into a pool melted using either arc or plasma discharge. Considerably smaller number of publications are devoted to using electron beam wire–feed additive manufacturing while one of its advantages of that it allows obtaining full fusion between the successively deposited layers. Another advantage is that the process is carried out in a vacuum chamber so that all adsorbed gases would be removed to eliminate any gas porosity. The third advantage is its high deposition rate.

The results of the investigation show that the composite alloys obtained in the course of this work contained no defects such as pores, thermal cracks or any other discontinuities; i.e., such a double-wire-feed additive process is suitable for obtaining fully dense samples and components. We believe this is a good result because the literature search shows that cracks appeared, for example, when fabricating a bimetallic component using such methods as (i) EBF^3^ deposition of a copper alloy on Inconel 625 [17] and (ii) laser-engineered net shaping on Inconel 718 [18]. Using the laser deposition from powder feedstocks also resulted in forming shrinkage porosity in a final copper–nickel alloy [4].

The microstructures obtained did not show any texturing which is a common finding in metals solidified under temperature gradient when high aspect ratio columnar grains are formed thus contributing to anisotropy of mechanical characteristics. In this case, one can see from Figure 2 that the microstructure is composed of almost equiaxed grains whose grain size distributions shown in Figure 3. 

Admixing melted Udimet-500 into an aluminum bronze melt pool resulted in the mutual dissolution of the alloy components so that β’-phase precipitation was suppressed because of aluminum being forced out of the copper matrix. Instead, up to ~8 at.% of Ni dissolves in copper-base solid solution while excess nickel forms NiAl either precipitates or grains depending upon the Udimet-500 content in the bronze. The heat of NiAl formation is ~58 kJ/mole, i.e., the maximum negative value as compared to those of other Ni_x_Al_y_ phases [19]. Therefore, NiAl formation is preferential to the melt of Udimet-500/aluminum bronze.

The composite alloy hardening is based not only on forming NiAl grains but also on forming both coarse and nanosized M_23_C_6_ carbides. The coarse ones look like carbide cores inside the NiAl shells while the nanosized ones precipitate inside the coarse NiAl grains.

Considering the fact that the limiting solubility of Cr, Co and Mo in Cu is low even in a liquid state [10] and analyzing the corresponding Cr-C and Cu-Ni-Al phase diagrams, it could be assumed that in the case of adding 5–25 vol.% of Udimet to aluminum bronze, high-temperature M_23_C_6_ carbide with a melting point at 1600 °C would be first to nucleate and grow in a melted-pool metal. Further solidification of the triple Cu-Ni-Al system is accompanied by precipitation of NiAl crystallites with some amount of dissolved Cu. It seems that M_23_C_6_ carbide may serve as an inoculation particle for the NiAl shell growth.

Precipitation of intermetallic compounds and carbides resulted in hardening the composite alloys (Figure 12) with the maximum hardness 2.6–2.8 GPa, tensile strength 650 MPa and 10% strain-to-fracture for the 25%Ud alloy. 

Fractography shows that the best distribution of hard phases was achieved in the case of 15% Ud which allowed retaining plasticity at ~20% with high enough tensile strength.

Both 5% Ud and 10% Ud composites contain fewer hard and brittle intermetallic compounds but more of the nickel–aluminum-bronze binder (Figure 14a,b). Such a microstructure makes them ductile enough (Figure 13) and plastically deformed when a ceramic ball slides over them, thus producing the glazed areas. Increasing the normal load leads to crushing the intermetallic phases and the generation of more wear particles plowing the glazed areas.

The 15% Ud composite contains more hard intermetallic phases, but these grains are homogeneously distributed in the matrix (Figure 14f). It seems that such a hardened structure still has some ductility to effectively resist the brittle fracture. In Figure 13, one can see that this composite still possesses ~20% ductility while its hardness is the second highest (Figure 12).

The hard-phase distribution in the hardest 25% Ud composite (Figure 14h) allows observing large agglomerated areas occupied by these phases. The ductility of this material is twice as low as that of the 15% Ud one (Figure 13). These brittle areas are subjected to fracture in sliding tests and thus easily produce hard wear particles, which then indent and scratch the surfaces. 

Almost all sliding wear dependencies in Figure 15a demonstrate that wear grows with the normal load except for that of the 5% Ud one which shows a plateau starting from a normal load of 14 N. Such a behavior may be typical with bronzes because of generating a tribological layer composing oxides and ductile bronze. Such a composite layer serves to increase the real contact area at higher loads and reduce adhesive wear component. It is rather typical behavior for bronzes in sliding contact and therefore they are used in sliding bearing. The worn surfaces of corresponding to different 14 N and 19N normal loads look all the same despite CoF reduction from 0.45 to 0.25. However, such a reduction is negligible from the viewpoint of tribology because it is all the same unlubricated adhesion sliding where CoFs as high as 0.5–0.1 occur. 

## 5. Conclusions

The electron-beam double-wire manufacturing of superalloy/aluminum bronze walls was carried out to study the effects of superalloy content on microstructure and phase formation. The effect of nickel dissolution in an aluminum bronze matrix resulted in the formation of a nickel–aluminum-bronze matrix with excess NiAl intermetallic compound precipitates and grains. In addition, primary eutectic M_23_C_6_ carbide structures were formed. NiAl grains contained fine M_23_C_6_ and nickel–aluminum bronze. Core–shell M_23_C_6_/NiAl particles were detected. The composites showed increased hardness and tensile strength but less plasticity. Sliding wear tests showed that wear increased with the normal load for all samples except for 15% Ud which possessed the optimal content and distribution of the hard phases in the bronze matrix. Such a microstructure allowed retaining about 20% of plasticity and, on the one hand, preventing its embrittlement while, on the other hand, increasing its strength and hardness against plastic flow wear.

## Figures and Tables

**Figure 1 materials-15-06270-f001:**
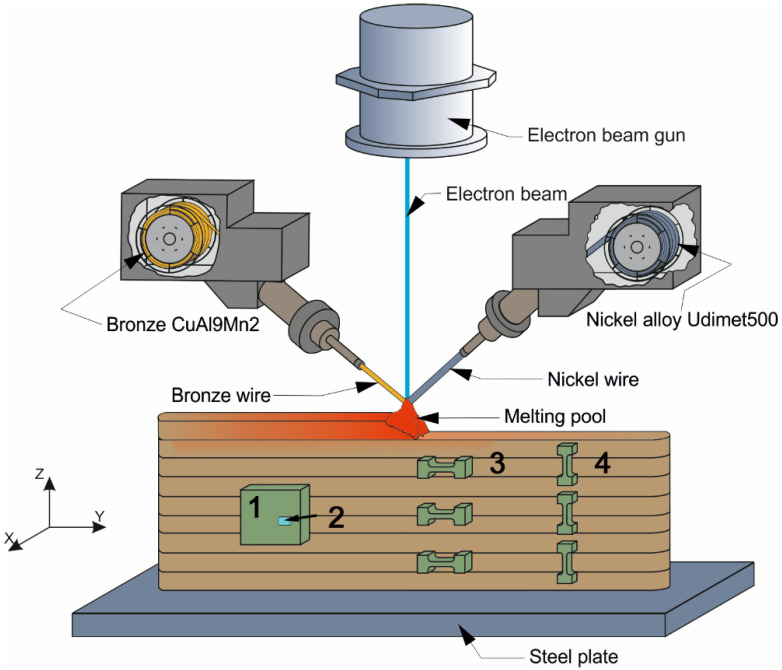
Schematic diagram of double-wire-feed electron-beam additive manufacturing. 1—samples for examination using optical microscopy, XRD and SEM; 2—thin foils for TEM; 3, 4—samples for mechanical testing.

**Figure 2 materials-15-06270-f002:**
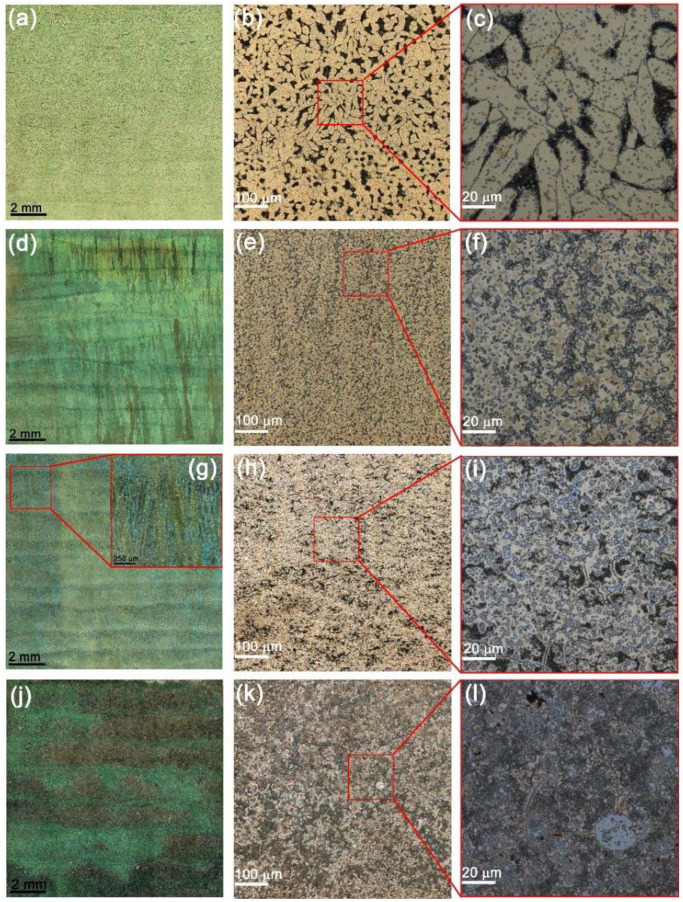
Microstructures of the 5% Ud (**a**–**c**), 10% Ud (**d**–**f**), 15% Ud (**g**–**i**) and 25% Ud (**j**–**l**) alloys.

**Figure 3 materials-15-06270-f003:**
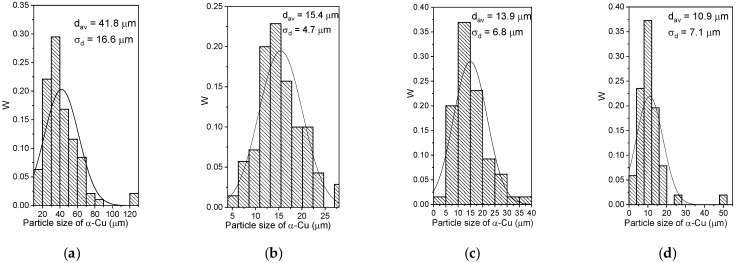
The matrix grain size distributions in 5% Ud (**a**), 10% Ud (**b**), 15% Ud (**c**) and 25% Ud (**d**) alloys.

**Figure 4 materials-15-06270-f004:**
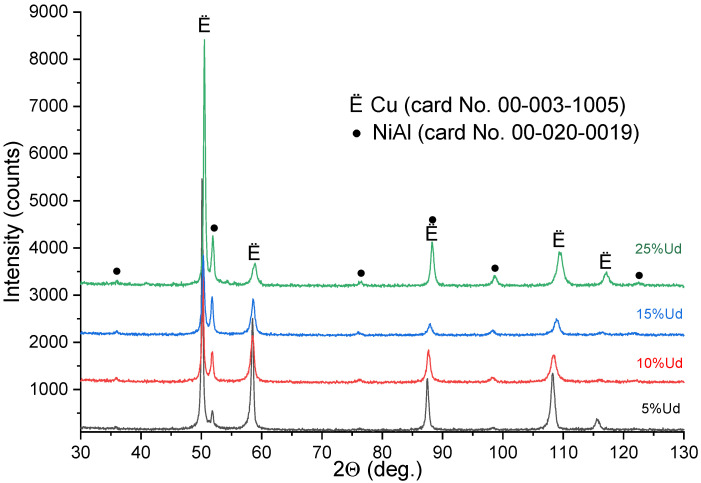
X-ray diffractograms of the bronze/Udimet-500 composites.

**Figure 5 materials-15-06270-f005:**
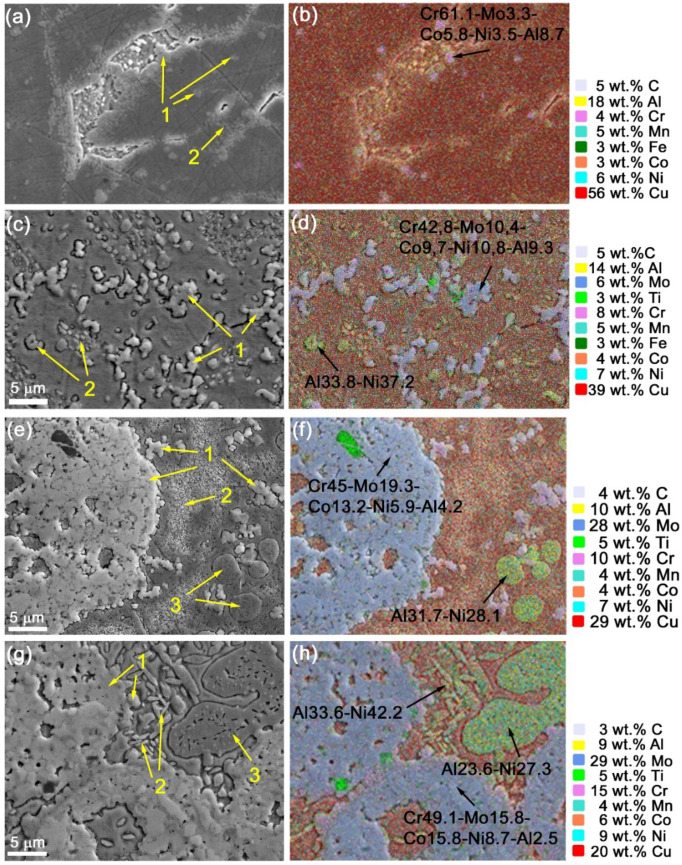
SEM images and EDS maps of the composite microstructures (**a**,**b**) 5% Ud; (**c**,**d**) 10% Ud; (**e**,**f**) 15% Ud; (**g**,**h**) 25% Ud. 1—Cr-rich particles; 2, 3—NiAl particles.

**Figure 6 materials-15-06270-f006:**
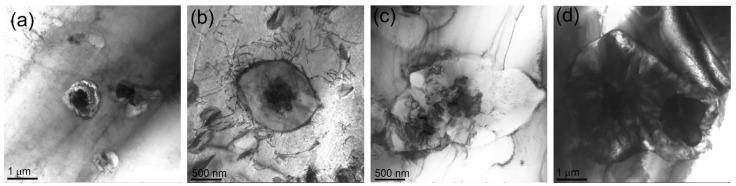
TEM bright-field images of Cr-rich particles in composites (**a**) 5% Ud, (**b**) 10% Ud, (**c**) 15% Ud and (**d**) 25% Ud.

**Figure 7 materials-15-06270-f007:**
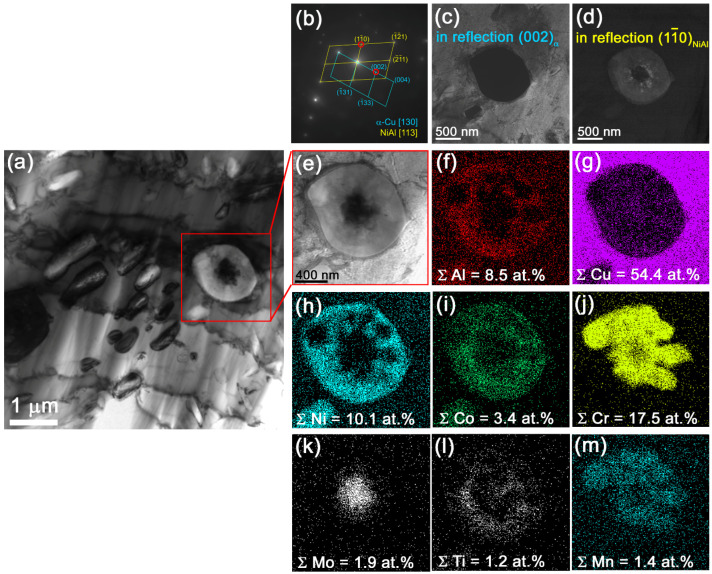
Bright-field TEM image of the 10% Ud composite (**a**,**e**), SAED pattern from a Cr-rich particle (**b**), dark-field images of the particle (**c**,**d**) in reflections (002)_α-Cu_ and (110)_NiAl_, EDS element distribution maps (**f**–**m**).

**Figure 8 materials-15-06270-f008:**
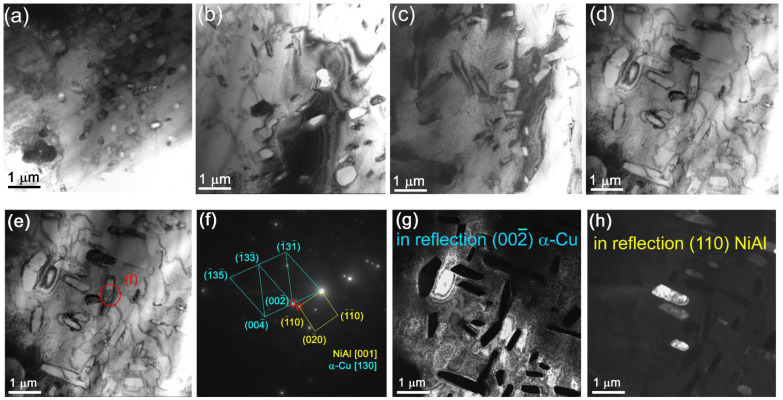
Bright-field TEM images of NiAl precipitates in composites: (**a**) 5% Ud, (**b**) 10% Ud, (**c**) 15% Ud and (**d**,**e**) 25% Ud. SAED pattern (**f**) from the area shown in (**e**), dark-field images (**g**,**h**) in reflections ${\left(00\bar{2}\right)}_{\alpha }$ and ${\left(110\right)}_{\mathrm{NiAl}}$.

**Figure 9 materials-15-06270-f009:**
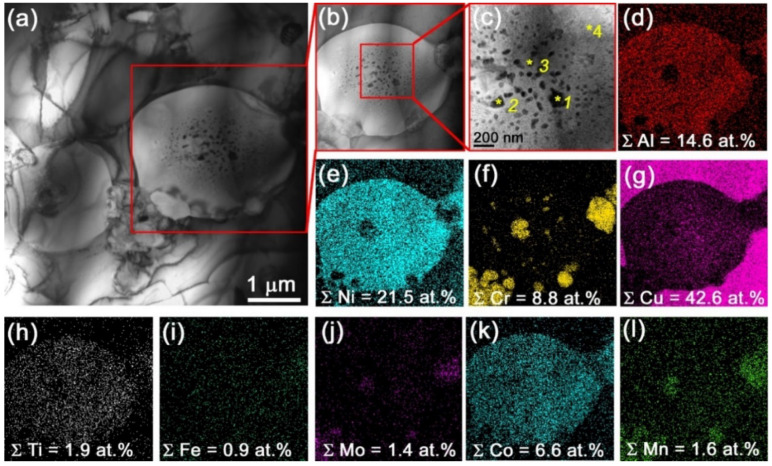
Bright-field TEM image of Ni-rich grains with fine precipitates in 15% Ud composite (**a**–**c**) and corresponding EDS element distribution maps (**d**–**l**). EDS element concentrations in zone 1 to 4 in (**c**).

**Figure 10 materials-15-06270-f010:**
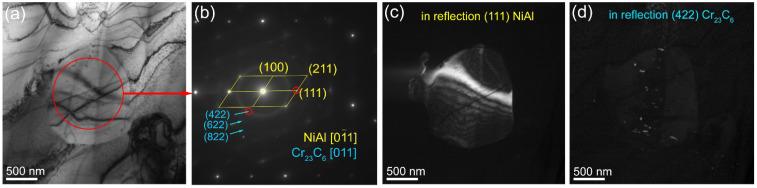
Bright-field TEM images of NiAl grain with carbide precipitates (**a**) in 15% Ud composite. SAED pattern (**b**) and dark-field images of NiAl grain (**c**) and M_23_C_6_ carbides (**d**) in reflections (111)_NiAl_ and (422)_Cr23M6_, respectively.

**Figure 11 materials-15-06270-f011:**
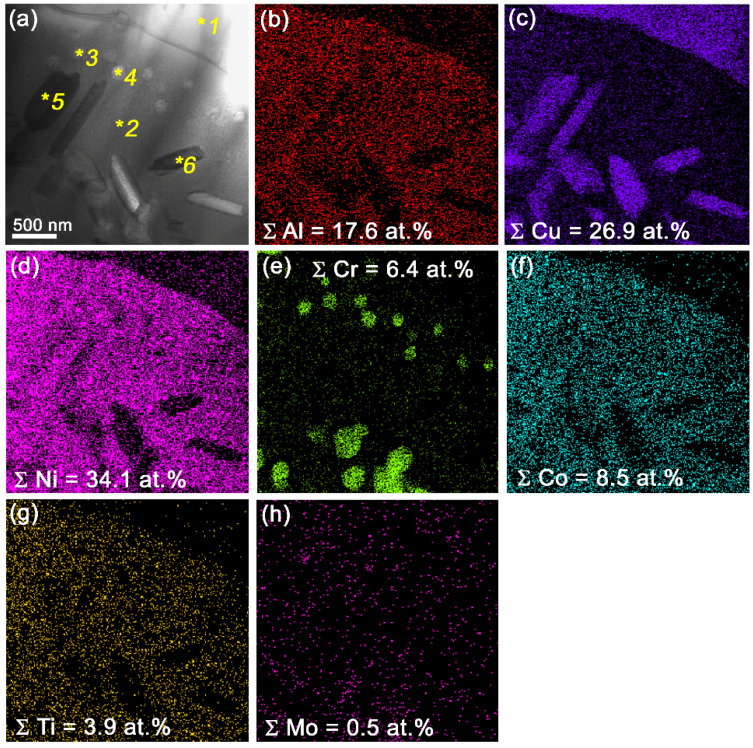
Bright-field TEM image of a NiAl grain in 25% Ud composite (**a**) with EDS element distribution maps (**b**–**h**) showing the presence of Cu-rich (**c**) and Cr-rich (**e**) particles. EDS element concentrations in zone 1 to 6 in (**a**).

**Figure 12 materials-15-06270-f012:**
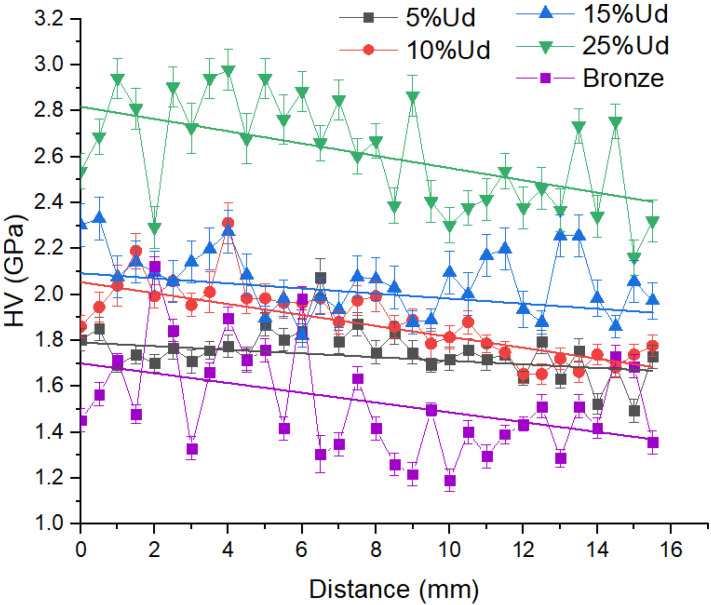
Microhardness profiles obtained along the Z axis.

**Figure 13 materials-15-06270-f013:**
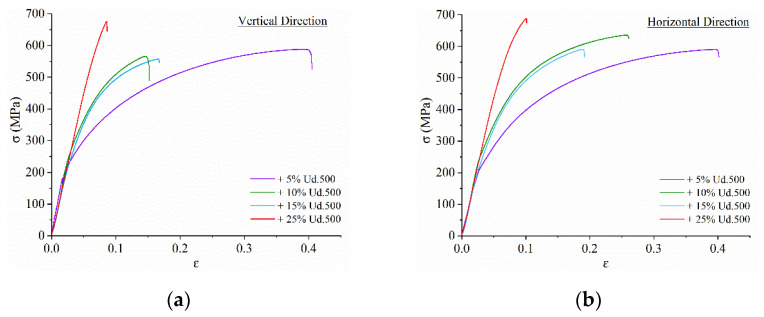
Stress–strain diagrams (**a,b**) and concentration dependencies of ultimate tensile stress and strain-to-fracture (**c**) for the composites.

**Figure 14 materials-15-06270-f014:**
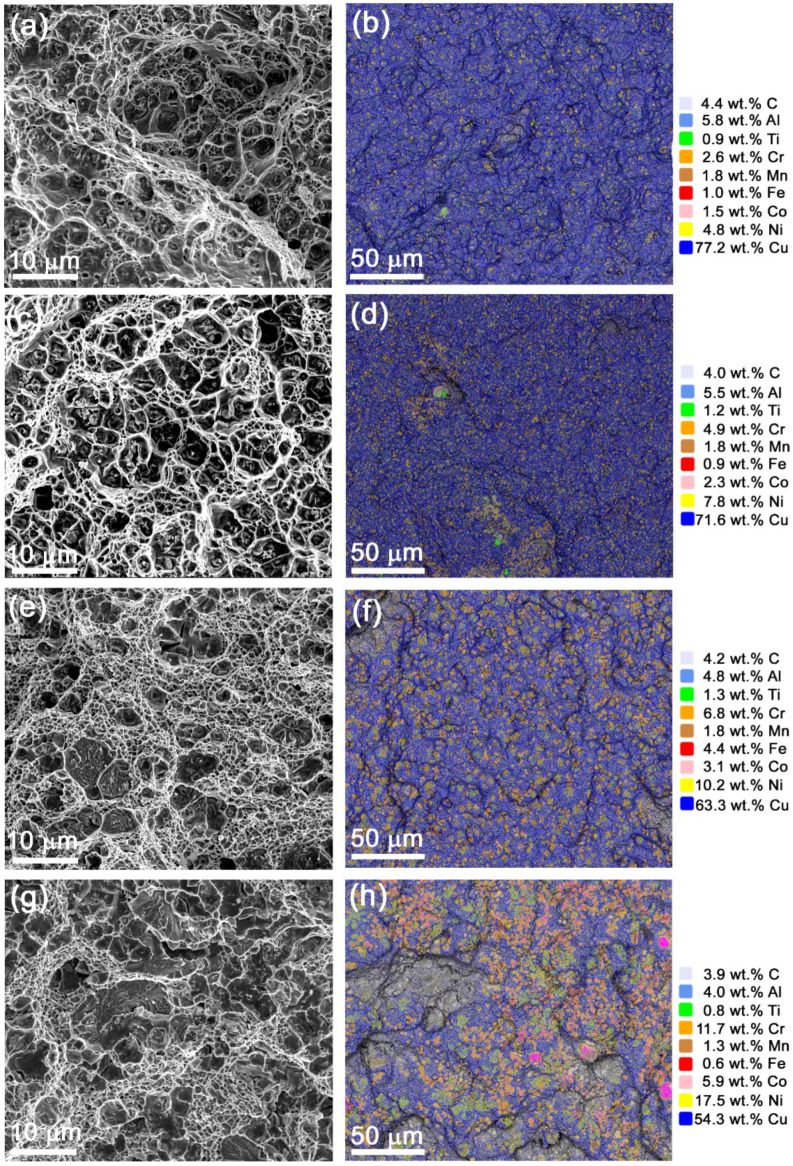
SEM SE fracture surface images of composites (**a**) 5% Ud, (**c**) 10% Ud, (**e**) 15% Ud and (**g**) 25% Ud and corresponding EDS element distribution maps (**b**,**d**,**f**,**h**).

**Figure 15 materials-15-06270-f015:**
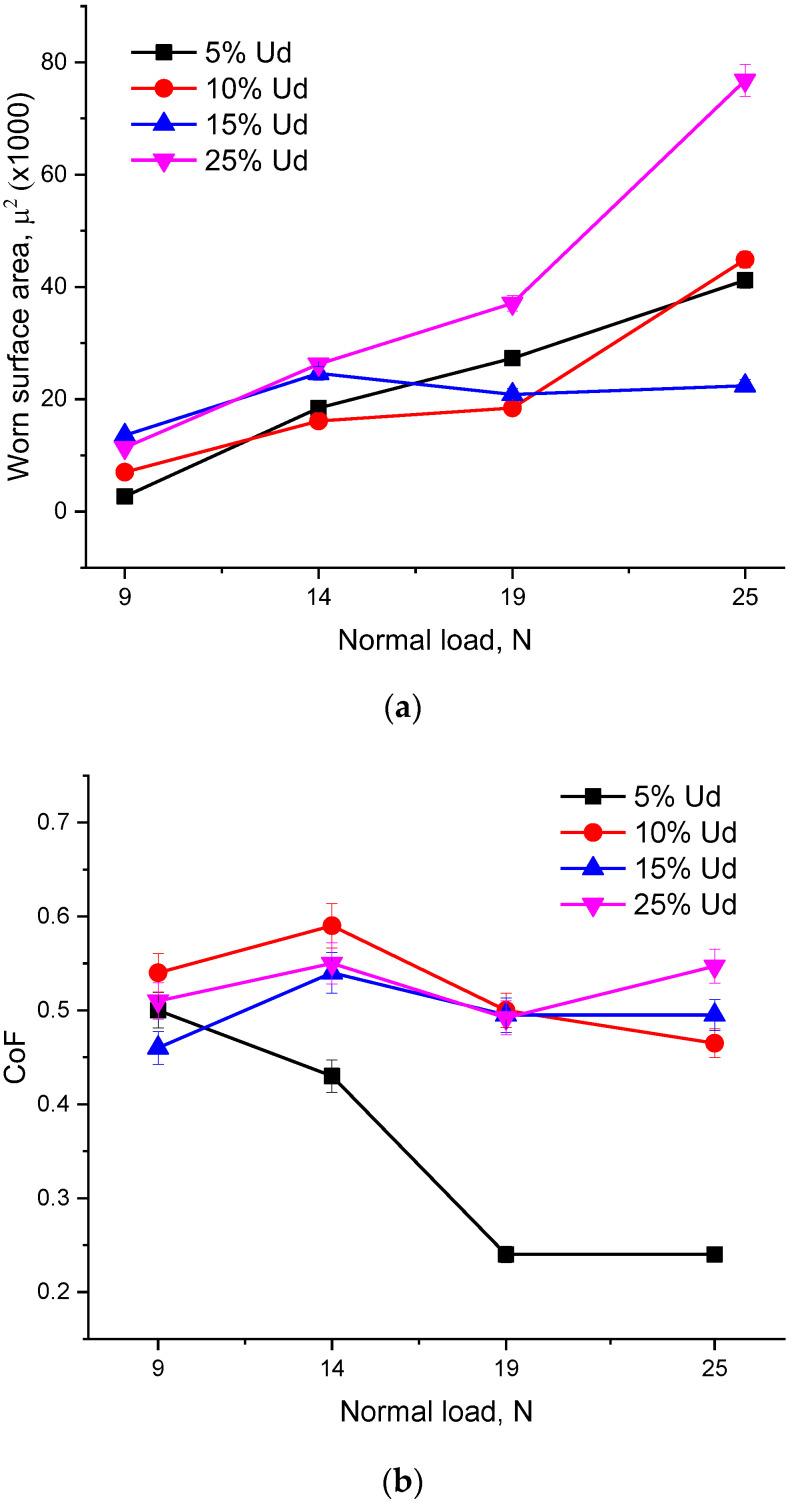
Wear (**a**) and coefficient of friction (CoF) (**b**) vs. normal load dependencies for the composite alloy samples.

**Figure 16 materials-15-06270-f016:**
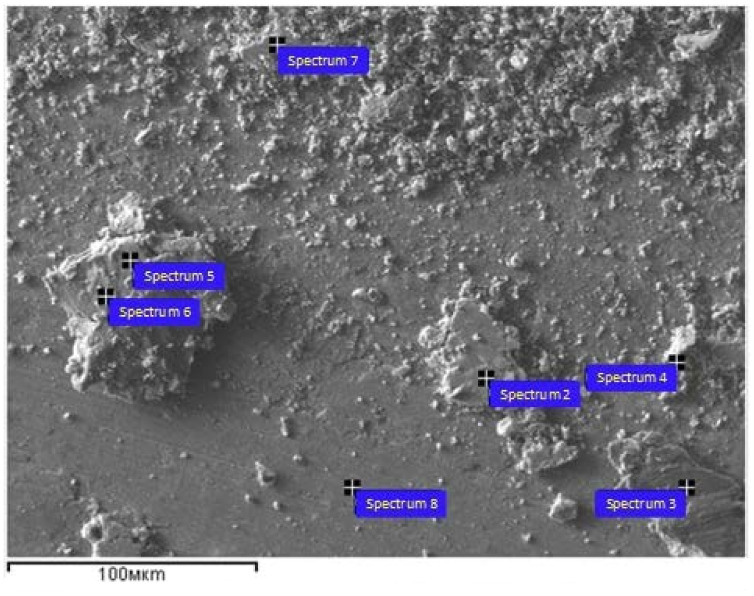
SEM image of worn surface obtained on the 15% Ud sample at 9 N normal load. 2–8 are the points where EDS spectra have been obtained (Table 4).

**Table 1 materials-15-06270-t001:** Chemical compositions of wires and CuAl9Mn2/Udimet-500 composites.

Material	Elements, wt.%											
	Al	Mn	Fe	Zn	Cu	Cr	Co	Mo	Ti	Si	Zr	Ni
CuAl9Mn2	9.3 ± 1.4	1.9 ± 0.04	0.3 ± 0.01	0.4 ± 0.02	bal.	-	-	-	-	-	-	-
Udimet-500	1.6 ± 0.2	0.1 ± 0.04	0.4 ± 0.02	0.04 ± 0.01	-	17.3 ± 0.1	13.2 ± 0.1	4.3 ± 0.03	2.7 ± 0.1	0.3 ± 0.03	0.04 ± 0.002	bal.
5% Ud	6.1 ± 0.6	1.5 ± 0.03	0.3 ± 0.01	0.3 ± 0.01	85.9 ± 0.6	0.9 ± 0.02	0.8 ± 0.01	0.3 ± 0.01	0.2 ± 0.01	-	-	3.6 ± 0.05
10% Ud	6.4 ± 0.5	1.4 ± 0.03	0.3 ± 0.01	0.2 ± 0.01	80.4 ± 0.4	2.1 ± 0.04	1.5 ± 0.02	0.5 ± 0.01	0.4 ± 0.02	-	-	6.7 ± 0.1
15% Ud	5.9 ± 0.5	1.4 ± 0.03	0.3 ± 0.01	0.1 ± 0.02	75.3 ± 0.4	3.2 ± 0.05	2.2 ± 0.03	0.8 ± 0.01	0.5 ± 0.02	-	-	10.1 ± 0.1
25% Ud	6.1 ± 0.6	1.1 ± 0.03	0.3 ± 0.01	-	64.1 ± 0.5	5.2 ± 0.06	3.7 ± 0.03	1.0 ± 0.02	0.9 ± 0.03	-	-	17.6 ± 0.1

**Table 2 materials-15-06270-t002:** EDS element concentrations in points indicated on the 15% Ud sample (Figure 9c).

Spectrum			Chemical Composition															
	Al		Ti		Cr		Mn		Fe		Co		Ni		Cu		Mo	
	wt.%	at.%	wt.%	at.%	wt.%	at.%	wt.%	at.%	wt.%	at.%	wt.%	at.%	wt.%	at.%	wt.%	at.%	wt.%	at.%
1	8.7	17.3	2.9	3.4	8.2	8.5	1.9	1.9	1.3	1.3	10.8	9.9	33.9	31.1	29.9	25.4	2.3	1.3
2	8.3	16.9	3.1	3.5	1.9	2.0	1.9	1.9	0.8	0.8	9.2	8.5	27.2	25.3	47.6	41.0	-	-
3	6.9	14.3	2.6	3.1	1.1	1.1	1.6	1.6	0.8	0.8	7.7	7.3	23.7	22.7	55.6	49.1	-	-
4	2.5	5.2	1.2	1.4	53.8	57.3	1.1	1.1	1.6	1.6	6.3	5.9	10.3	9.7	14.9	13.0	8.2	4.7

**Table 3 materials-15-06270-t003:** EDS element concentrations in points indicated on the 25% Ud sample (Figure 11a).

Spectrum			Chemical Composition															
	Al		Ti		Cr		Mn		Fe		Co		Ni		Cu		Mo	
	wt.%	at.%	wt.%	at.%	wt.%	at.%	wt.%	at.%	wt.%	at.%	wt.%	at.%	wt.%	at.%	wt.%	at.%	wt.%	at.%
1	2.6	5.8	-	-	-	-	1.0	1.1	0.4	0.5	0.9	0.9	7.5	7.7	87.6	83.9	-	-
2	11.6	22.1	5.1	5.5	2.2	2.3	1.7	1.6	0.8	0.7	12.3	10.7	50.5	44.3	15.9	12.9	-	-
3	9.3	17.6	2.7	2.9	37.7	37.1	1.3	1.2	0.9	0.9	8.0	7.0	28.2	24.5	9.6	7.7	2.3	1.1
4	3.1	6.0	1.3	1.4	71.3	72.5	-	-	0.9	0.8	5.8	5.2	8.5	7.7	4.9	4.1	4.2	2.3
5	6.4	13.5	1.9	2.2	1.2	1.3	1.0	1.1	0.6	0.6	5.8	5.6	24.9	24.1	58.2	51.8	-	-
6	4.3	9.3	1.8	2.1	0.9	1.1	0.9	1.0	0.6	0.7	4.7	4.7	20.6	20.4	66.1	60.7	-	-

**Table 4 materials-15-06270-t004:** EDS element concentrations in points indicated on the 15% Ud sample (Figure 16).

Spectrum	Chemical Composition													
	O		Al		Cr		Mn		Co		Ni		Cu	
	wt.%	at.%	wt.%	at.%	wt.%	at.%	wt.%	at.%	wt.%	at.%	wt.%	at.%	wt.%	at.%
2	8.0	23.8	7.1	12.5	0.2	0.2	1.4	1.2	0.4	0.4	2.9	2.4	79.9	59.6
3	5.1	16.1	7.6	14.2	1.0	0.9	1.4	1.2	0.3	0.3	2.8	2.4	81.8	64.8
4	-	-	0.3	0.7	0.8	0.9	1.8	2.1	0.6	0.6	2.8	4.2	93.5	94.5
5	0.9	3.0	6.6	13.8	0.2	0.2	1.4	1.5	0.2	0.2	3.0	2.9	87.8	78.4
6	2.6	8.7	7.2	14.4	0.7	0.8	1.7	1.7	0.7	0.7	3.9	3.6	83.1	70.2
7	9.6	27.6	6.5	11.1	0.5	0.4	1.3	1.1	0.5	0.4	3.4	2.7	78.2	56.7
8	0.6	2.2	6.9	14.7	0.3	0.4	1.3	1.4	0.3	0.3	2.7	2.6	87.8	78.5

## Data Availability

Data sharing is not applicable to this article.

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
