# Peer review of "Aluminum Bronze/Udimet 500 Composites Prepared by Electron-Beam Additive Double-Wire-Feed Manufacturing"

_materials, 2022, doi:10.3390/ma15186270_

Round 1
Reviewer 1 Report
The present manuscript reports an interesting investigation on the fabrication of aluminum bronze/Udimet 500 composites with electron beam additive double wire feed manufacturing. Microstructure and mechanical properties such as hardness, tensile strength and ductility have been characterized. It is an interesting topic suitable for publication in Materials. However, more detailed discussion and revision should be made to improve the quality of the present work.
1. Extensive experimental studies have been carried out on the additive manufacturing of nickel-base superalloys with approaches such as selective laser melting, laser metal deposition and selective electron beam melting (Acta Mater. 2012, 60, 2229; J. Iron Steel Res. Int. 2022, 29, 1322). In comparison with these approaches, what is the advantage of the current fabrication approach? Have the current fabrication approach been used in literature?
2. Reading the present manuscript, it is not necessarily clear to the reader where the novelty of the work lies.
3. There are some typos in the manuscript. For example, 5%Ud in line 72 denotes 5 vol.% of Udimet-500, while 5%Ud in line 104 denotes 5 wt.% of Udimet-500. Figure 17 in line 190 should be Figure 12. The description in line 192 is in conflict with the data in Figure 12.
4. The authors characterized the microstructure of the CuA19Mn2/Udimet500 alloy. However, the effect of microstructure on mechanical properties of the alloy has not been discussed.
5. Due to the precipitation of strengthening phase, the increase of strength is often accompanied by the decrease of plasticity. The authors should provide stress–strain curves of the fabricated samples are required, and discussion how they depend on the microstructure.
6. In addition to microstructure, it is better to add discussion on how crystallographic texture affect mechanical properties of the fabricated samples.
Author Response
Please, see the file attached

Reviewer 2 Report
A very interesting paper. I myself also think that composite and in-situ alloy production with additive manufacturing technologies is one of the great potentials for the future. I have only one minor suggestion for your paper. You only wrote a really short description of how the samples were manufactured. You mention which beam current and wire feed rate was used. I am really curious how did you got these parameters. Surely some trials with different parameters were done, before you got good samples. Because you also mention in the paper that your samples contained no defects while other researchers made samples with cracks and other defects. After adding some explanation about how you manufactured good samples, I will have no problem with recommending this paper for publication.
Author Response
Please, see the file attached

Reviewer 3 Report
The development of composite materials alloy by double wire electron beam bonding of Udimet-500 and bronze has been investigated and valuable results have been obtained. However, the following issues should be resolved:
(1) While wear resistance is important, data on the condition of worn surfaces and worn particles is not provided and discussed. It should be added.
(2) The reason why the coefficient of friction decreases at high loads in the case of 5%Ud should be properly discussed.
(3) Data and discussion on microstructure and adhesion at the boundaries between adjacent layers should be added.
(4) Title of Fig. 14: (d) 15%Ud => (c) 15%Ud
Author Response
Please, see the file attached

Round 2
Reviewer 1 Report
The manuscript is ready for publication.
Reviewer 3 Report
I've understood your answers for my comments. They are almost acceptable.